# Numerical Modeling of the Effects of Toe Configuration on Throughflow in Rockfill Dams

Nils Solheim Smith [1,*], Ganesh H. R. Ravindra [1,2] and Fjóla Guðrún Sigtryggsdóttir [1]

1. Department of Civil and Environmental Engineering, Norwegian University of Science and Technology (NTNU), S.P. Andersens veg 5, 7491 Trondheim, Norway; ganesh.hiriyanna.rao.ravindra@trondheim.kommune.no (G.H.R.R.); fjola.g.sigtryggsdottir@ntnu.no (F.G.S.)
2. Department of Water and Wastewater, Kommunalteknikk, Trondheim Kommune, Erling Skakkes Gate 14, 7013 Trondheim, Norway
* Correspondence: nils.solheim.smith@ntnu.no

**Abstract:** The rockfill toe structure situated within the downstream slope of rockfill dams is an integral part of a defense mechanism safeguarding the dam structure in throughflow situations. Recent studies have concluded that the rockfill toe structure can have significant impacts on throughflow development and stability of rockfill dams under scenarios of accidental throughflow caused by overtopping of the dam core. The ability to numerically model the effect of various toe configurations on flow through rockfill dams can support the design of effective toe drainage structures for rockfill dams. Development and calibration of a reliable numerical modeling tool in this regard has been challenging owing to lack of availability of extensive datasets from physical modeling investigations. This study further employs datasets gathered by a recent physical modeling study investigating the effects of various toe configurations on throughflow development in rockfill dam models. A commercial numerical seepage modeling tool with an option for non-Darcy flow was calibrated against the datasets with good calibration metrics. The study is novel in providing a rare report on the usage of this option. The calibrated tool can further be employed to carry out a wide array of simulations to arrive at an ideal design for a toe structure for rockfill dams and for assessment of hydraulic performance of toe structures.

**Keywords:** rockfill dams; dam safety; throughflow; numerical modeling; non-Darcy flow

## 1. Introduction

Embankment dams, constructed with locally excavated earth or rockfill represent 78% of the total number of existing dams worldwide [1]. Embankment dams comprising of coarse rockfill materials in more than 50% of the dam volume are defined as rockfill dams and represent 13% of the worldwide dam population [1]. A rockfill dam structure generally consists of an impervious element, filter zones, support fill and some means of controlling the development of phreatic surface and seepage through the dam structure.

The issue of dam safety has gained much attention in the recent past. Stringent measures are being put in place by the respective national dam safety authorities to ensure safety of dams. Although there exists significant amount of accumulated scientific literature within the research discipline of embankment dams in general, technical literature describing throughflow behavior of rockfill dams is scarce. This article aims at adding to the research discipline of rockfill dam safety. Dam safety assessment is a complex task, as it is influenced by multitudes of internal and external factors [2]. It is essential to determine the most common causes of dam failures to identify probable factors which commonly contribute to dam instability. Statistics from the International Commission on Large Dams (ICOLD) state overtopping as the main cause of embankment dam failure appearing as the primary factor in 31% of the total number of failures, and is further involved in another 18% of failures as a secondary agent [3]. Hence, equipping embankment dams with defense

mechanisms against unanticipated overtopping or leakage events is of importance from a dam safety perspective. This includes safeguarding against accidental throughflow conditions arising when the core of a rockfill dam is overtopped, resulting in turbulent flow within the downstream dam shoulder, as shown in Figure 1b.

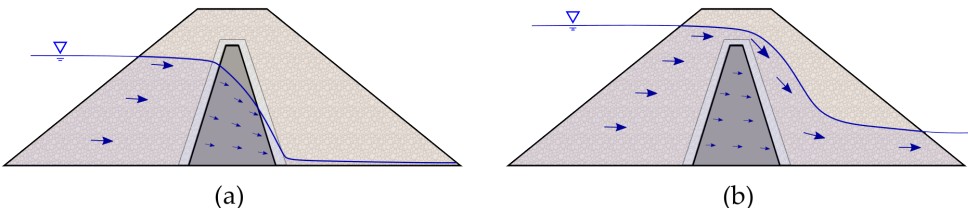

|  |  |
|:-:|:-:|
| (a) | (b) |

**Figure 1.** Sketch of throughflow situations for an embankment dam with a central core. (**a**) Normal conditions with seepage through the core. (**b**) Accidental load situation with overtopping of the core leading to large throughflow.

The rockfill toe structure situated within the downstream slope of rockfill dams can be considered as an integral part of a defense mechanism installed to protect the dam structure under normal seepage situation as described by Figure 1a, as well as under accidental overtopping situations leading to extreme throughflow conditions as shown in Figure 1b. In fact, some previous studies into embankment dam failures describe the downstream toe as a critical location for failure initiation under throughflow scenarios [4–8]. Furthermore, findings of Toledo and Morera [9] and Moran and Toledo [10] suggest that rockfill toes may be used as effective protection against throughflow in rockfill dams. Furthermore, Moran et al. [11] present a procedure for the design of external toe protection for rockfill embankments.

A recent investigation conducted by Ravindra [12] studied the effects of various configurations of rockfill toes on throughflow development within hydraulic scale models of rockfill dams (Figure 2). Findings from the experimental studies, presented by Kiplesund et al. [13], highlight the fact that toe configuration can have significant impact on the development and progression of phreatic surfaces within rockfill dam models subjected to incremental overtopping scenarios. The toe configurations were also found to influence the stability of the downstream slope. The study gave valuable insight into the significance of rockfill toes with regard to rockfill dam safety. The present study builds numerical models on the data accumulated through the experimental investigations [12,13].

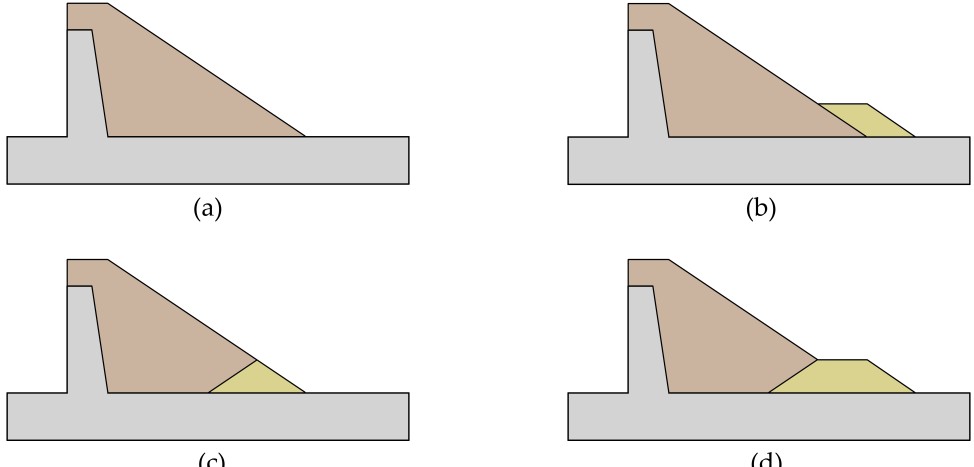

|  |  |
|:-:|:-:|
| (a) | (b) |
| (c) | (d) |

**Figure 2.** Displaying the investigated toe configurations, where (**a**) shows no toe configuration, (**b**) external toe, (**c**) internal toe and (**d**) combined toe configuration.

The present study is a part of an ongoing research program into dam overtopping and throughflow. Figure 3 visualizes the modeling strategies of this overarching research

program from full-scale to model-scale investigations, as well as numerical modeling efforts. The full-scale part embraces consideration of the design of existing dams [14,15], as well as analysis of data accumulated from full-scale tests [16]. The hydraulic scale models that relate to the numerical modeling of the present study are those presented by Kiplesund et al. [13]. These considered the scaling of previous hydraulic scale models for investigating riprap erosion protection on the downstream slope of embankment dams [17–19]. The combined application of the different modeling strategies is for enhanced applicability and relevance of the research for full-scale dam cases.

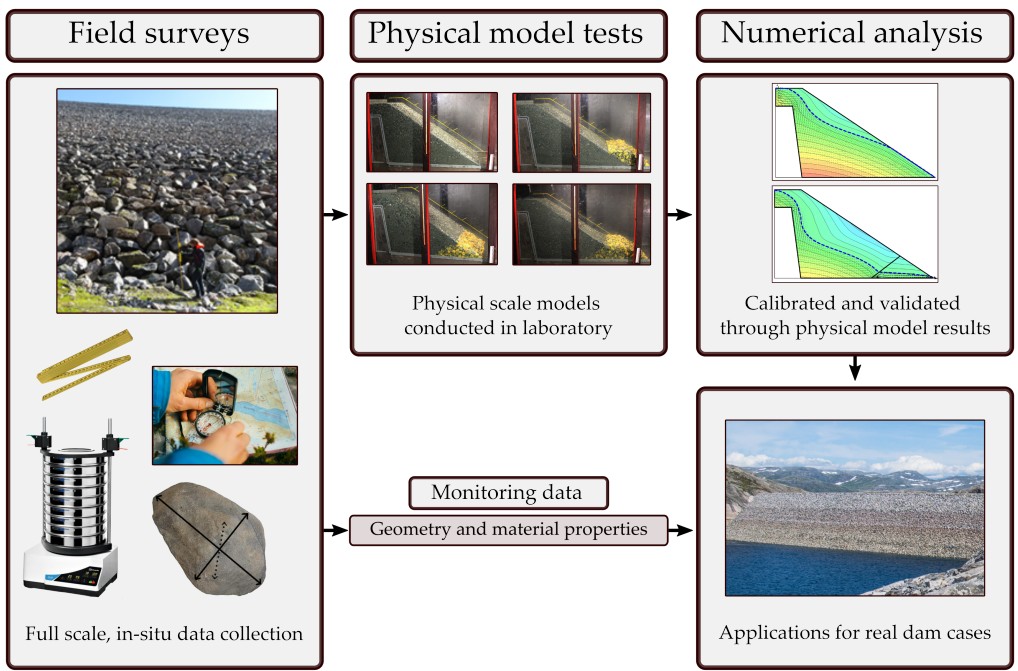

**Figure 3.** Schematic overview of the overarching research program.

The overarching goal of the present part of the research program is to evaluate the hydraulic response of rockfill dams exposed to accidental throughflow scenarios (Figure 1b) and to study the effects of rockfill toes on throughflow hydraulic properties of rockfill dams. For this purpose, a numerical model is developed for replicating results from physical model tests considering turbulence of the flow. Hence, an important aspect of the present study is the implementation of a geotechnical software [20] commonly employed in dam engineering for practical applications as well as in research [21,22]. However, the modeling usually assumes laminar flow or Darcy flow conditions, suitable for cases as in Figure 1a. Thus, the present study aims at investigating the ability of a tool provided within such software [23] to model turbulent or non-Darcy flow commonly encountered in the physical rockfill dam models. This has a relevance when proceeding to numerical models of real dam cases considering non-Darcy flow for the accidental overtopping situation. Moreover, numerical modeling of the effect of various toe configurations on flow through rockfill dams has not been looked into in the past. The datasets gathered through the previously mentioned physical modeling investigations [12,13] are used to calibrate numerical models employing the numerical seepage software SEEP/W [20] with a non-Darcy tool. The aim is to predict the development of throughflow within rockfill dam structures and to numerically model the effect of a drainage component within the downstream dam slope on non-linear throughflow development.

## 2. Background

Flow through porous media is generally characterized as either Darcy or non-Darcy type based on flow properties. The linear Darcy flow theory is widely implemented in soil mechanics and is described by the following equation:

$$v = k\,i, \tag{1}$$

where the velocity of flow, $v$, is described by a linear relationship between hydraulic conductivity, $k$, and the hydraulic gradient, $i$.

Darcy's law is only valid at low velocities, i.e., laminar flow. At higher velocities, the inertial forces distort the streamlines and turbulent flow occurs, removing the linear relationship. In rockfill material, the voids are of a magnitude that turbulent flow is expected [16]. An illustration is made in Figure 4. Where Darcy's law is applicable, flow is evenly distributed and laminar. In the rockfill case, the voids are bigger and velocities vary along with the grains redirecting the flow.

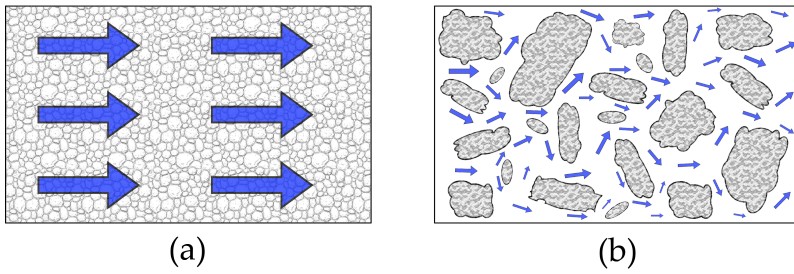

(a)                                         (b)

**Figure 4.** Representation of different flow regimes through porous media. Where (**a**) shows laminar flow through uniform small-grained material and (**b**) demonstrates turbulent flow condition through material with coarser grains and larger voids.

Non-Darcian or turbulent flow through porous media is generally represented as a power-law function:

$$v = a\,i^{b}, \tag{2}$$

where, $a$ and $b$ represent empirical coefficients to be determined experimentally. Coefficient $a$ depends on the properties of fluid and porous media such as porosity, particle shape, particle size, roughness, tortuosity of void structure and viscosity of fluid. Parameter $b$ is dependent upon the state of flow or the level of flow turbulence [8].

Until recently, the performance of the general non-linear flow law of the form presented in Equation (2) was verified only through experimental studies conducted in permeameters. Past studies have investigated non-linear flow through rockfill medium through elaborate permeameter experimental testing conducted on rockfill with sizes ranging from $d_{50} = 10$ mm to 240 mm [8,24–29]. Several empirical relationships describing the non-linear $i$-$v$ flow properties have also been proposed as a result of these investigations. Although a considerable number of investigations have investigated non-Darcian flow through rockfill material in permeameters of varying sizes, experimental validation of these past findings in rockfill embankments exposed to throughflow conditions can be stated as quintessential for validation of past research findings in terms of relevance of application in rockfill dam engineering. A recent study put forth by Ravindra et al. [16] has made attempts at validating some of the widely employed non-linear flow equations from the past and have also further proposed a new equation applicable for non-linear flow through homogeneous rockfill dams.

Dealing with soil or rockfill, which is generally heterogeneous and discontinuous in nature, approximate solutions are normally pursued [30]. The finite element method is a powerful tool for approximating complex field problems. The domain in which the analysis is being conducted is divided into finite elements creating a mesh. For each node in the mesh, the field variable is explicitly calculated through a mass balance approach.

The functions that define how the field variable varies in the domain are controlled through the material properties. The mass balance approach for the utilized software relevant for this study can be summarized by the following general equation [20]:

$$\frac{dM_{st}}{dt} = \dot{m}_{in} - \dot{m}_{out} + \dot{M}_S \tag{3}$$

where $M_{st}$ is the stored mass in the control volume, the inflow and outflow terms, $\dot{m}_{in}$ and $\dot{m}_{out}$, represent flow in and out of the control volume and $M_S$ is the source term, with dot-notation representing rates.

For seepage problems, the governing differential equation utilized by the software in a 2D case is defined by:

$$\frac{\partial}{\partial x}\left(k_x \frac{\partial H}{\partial x}\right) + \frac{\partial}{\partial y}\left(k_y \frac{\partial H}{\partial y}\right) + Q = \frac{\partial \theta}{\partial t} \tag{4}$$

where $k$ is the hydraulic conductivity in x- and y-direction, $H$ is the total head, $Q$ is the mass source or sink term. The right side of the equation is the change in volumetric water content, $\theta$, with respect to time. The simulations in this study are conducted in a steady-state condition, which yields that there are no time-dependant variables and the right side of Equation (4) becomes zero.

The finite element method then entails that the governing differential equation must be satisfied at every node; assembled in matrix form, this can be summarized as the finite element equation:

$$[\mathbf{K}]\{\mathbf{h}\} = \{\mathbf{q}\} \tag{5}$$

where $\mathbf{K}$ represents the global element matrix, defining each element's geometry and material properties; $\mathbf{h}$ is the primary unknown vector consisting of the total head at each node. Lastly, $\mathbf{q}$ is the resultant vector also called the nodal flow vector, defined by the boundary conditions. This system of equations is iteratively solved so that each element in addition to the whole domain satisfies the governing equation.

To account for non-linearity of the flow, or non-Darcy flow, an added feature is usually required. Professional packages are available that employ a flux approach where the nonlinear nature of Equation (2) is relegated to an apparent hydraulic conductivity term, $k_{w,a}$, by rearranging the equations as follows with the hydraulic gradient expressed in vector notation as $\nabla h = \left(\frac{\partial h}{\partial x}, \frac{\partial h}{\partial y}, \frac{\partial h}{\partial z}\right)$ and velocity expressed as a flux vector, $q_w$ [23]:

$$q_w = -k_{w,a}\,\nabla h \tag{6}$$

where the apparent hydraulic conductivity, $k_{w,a}$ can be expressed in terms of total head as [23]:

$$k_{w,a} = \frac{-1 + \sqrt{1 + 4\,C_F\,k^{3/2}\sqrt{\frac{\rho_w}{g\mu_w}}\,|\nabla h|}}{2\,C_F\sqrt{k}\sqrt{\frac{\rho_w}{g\mu_w}}|\nabla h|} \tag{7}$$

where $C_F$ is the form drag constant, $k$ is the hydraulic conductivity of the porous media, $\rho_w$ is fluid density, $g$ is gravitational acceleration and $\mu_w$ is the dynamic viscosity of fluid.

It can be seen from Equation (7) that the apparent conductivity will decrease with increasing velocity, providing non-linear behavior of the rockfill material. Definition of the parameters that govern the non-linear behavior are based on the closed form equation for hydraulic conductivity derived by Van Genuchten [31]. The input parameters include, saturated and residual water content, $\alpha$, $n$, $l$, $C_F$ and fluid temperature. Originally developed for agrophysical purposes, the equation builds on the soil water retention curve, which can be established through laboratory testing, to find the relative conductivity between saturated and unsaturated material.

Listing some parameters found for clay to sandstone soils, $n$-values ranges from 1.2 to 10 [31]. In a later study, typical values are presented as 1.2 for fine soils and 2.7 for coarse soil [32]. For the $\alpha$-parameter variation lies between 0.01 and 1 for fine material including clay [33,34]. There exist multiple studies with varying values for the form drag constant, $C_F$, and there are no input limitations in the add-in of the software used [20]. As a selected limitation for the present study, the drag constant can vary between 0.5 and 1.5 for coarse granular material [35]. The $l$-parameter represents the inter connectivity and tortuosity of the voids in the material, with values ranging from $-1$ to 2 in different solutions [32].

Several past studies available in the international literature perform the function of defining turbulent flow through uniform and homogeneous rockfill materials [8,16,24–29]. However, the validity of these equations as applied to zoned rockfill structures comprising several different materials with varying properties has not been investigated. This can be attributed to the fact that hydraulic throughflow properties in zoned rockfill dam models can be very complex and deriving general results/relationships to describe such behavior can be challenging. Hence, numerical modeling can be considered as a well suited method for investigating such complex hydraulic aspects in rockfill dams. This study aims at employing a numerical model to obtain a representative description of flow through rockfill dam models with two individual zones. This can form a strong launchpad for further developments to the model which can help improve our capabilities to model complex hydraulic behaviors within large scale rockfill dams.

## 3. Materials and Methods

Methodology, instrumentation and material properties adopted for the physical modeling studies are succinctly explained in the following chapter. The process relating to the numerical analysis is then explained, covering both the design of the model and analysis.

### 3.1. Physical Model

The physical models and results are described in detail by Kiplesund et al. [13] and only the main features are presented herein. The rockfill dam models (Figure 5) were built in a 25 m long, 1 m wide and 2 m high flume at the hydraulics laboratory of NTNU, Trondheim. The effects of various configurations of rockfill toes (no toe, internal toe, external toe and combined toe configurations) on throughflow development within rockfill dam models were studied. The model only consisted of the downstream half of a rockfill dam structure with an aluminum core built on a horizontal platform elevated 350 mm from the flume floor. The reasoning being that under throughflow conditions, behavior of the downstream shoulder is of specific interest from a dam safety standpoint in comparison to the upstream embankment as the downstream slope of rockfill embankments are exposed to higher degree of destabilizing forces under turbulent throughflow conditions.

The physical model of a rockfill dam with a toe structure is presented in Figure 5. Region (a) within Figure 5 represents the internal toe configuration, Region (b) the external configuration and Regions (a) + (b) represent the combined configuration. The individual setups are visualized in Figure 2. The instrumentation for the physical model is comprised of 8 pressure sensors installed along the dam foundation (P3–P10 seen in Figure 5) for measurements of the pore pressure distributions within the dam models under throughflow conditions. The pressure measurements in this study are shown as the piezometric head, using the origin in Figure 5 as the datum. Additional pressure sensors were installed (i) on top of the metallic dam core (P2) and (ii) at the upstream section of the model (P1) for measurements of water levels over the core and the upstream reach of the model, respectively. Discharge to the flume was fed by two pumps with a combined capacity of about 0.4 m$^3$/s regulated through a digital discharge meter. The physical tests were conducted so that the discharge was fixed and the resulting water level upstream of the dam variable.

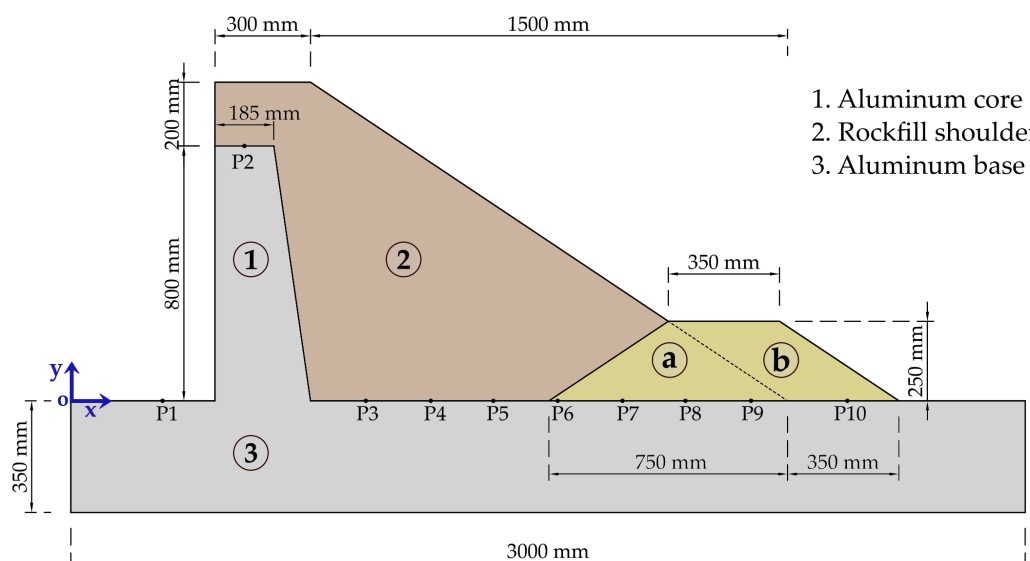

**Figure 5.** Sketch of the model displaying the rockfill shell, along with the base and core, with two regions for (**a**) internal and (**b**) external toe configuration. The coordinate system, drawn in blue, is placed at the origin. The locations of the installed pressure sensors are listed as P1–P10 (see Table 1).

**Table 1.** Sensor position along the dam body, represented as the *x*-axis in Figure 5.

| Sensor | P1 | P2 | P3 | P4 | P5 | P6 | P7 | P8 | P9 | P10 |
|---|---|---|---|---|---|---|---|---|---|---|
| Position (m) | 0.29 | 0.54 | 0.93 | 1.13 | 1.33 | 1.53 | 1.73 | 1.93 | 2.14 | 2.44 |

The selected rockfill material grain size was based on data analysis from existing rockfill dams in Norway. The gradation curves were down-scaled by a ratio of 1:10, barring some of the finest materials due to limitations of the flume pumping system. The selected gradation curve was thus slightly narrower, and lies on the coarser boundary of Norwegian standards. In total, 1800 kg of shell material was mixed in order to complete the model. Some key material parameters for the rockfill shoulder and toe materials are presented in Table 2. Presented are density, key grain sizes and the coefficient of conformity, $c_u = d_{60}/d_{10}$. The resulting gradation curves for the well-graded shell material and uniform toe material can be seen in Figure 6.

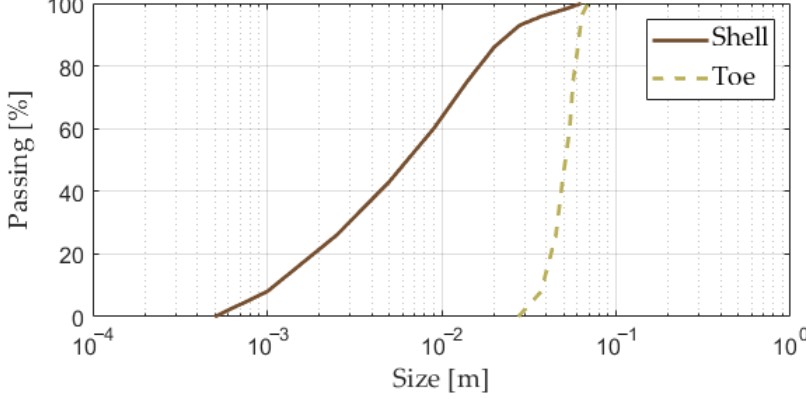

**Figure 6.** Gradation curve shown for shell and toe material.

**Table 2.** Material properties for shell and toe.

|  | $\rho$ | $d_{10}$ | $d_{50}$ | $d_{60}$ | $c_u$ |
|---|---|---|---|---|---|
| **Shell** | 2720 | 1.2 | 6.5 | 9.0 | 7.50 |
| **Toe** | 2860 | 37 | 52 | 55 | 1.42 |
|  | (kg/ m$^3$) | (mm) | (mm) | (mm) | ( - ) |

A total of twelve physical tests were conducted, comprising three tests on each individual toe configuration. The testing methodology consisted of exposure of the rockfill dam models to incremental throughflow magnitudes. The discharge intervals were adopted as $\Delta q = 0.5 \times 10^{-3}$ m$^3$/s over $N$ discharge steps with initial exposure set to $q_i = 1 \times 10^{-3}$ m$^3$/s. The discharge levels were maintained constant over regular time periods of $\Delta t = 1800$ s to allow for flow stabilization at each overtopping interval. Reference is made to Kiplesund et al. [13] for detailed summary of the tests and the respective testing protocols. The tests were visually documented through video recordings.

*3.2. Numerical Model*

Independent of which software one is interested in utilising, the modeling procedure begins with defining the domain to investigate, i.e., drawing the geometry. This can be imported from CAD software or can be defined within the selected software for numerical analysis. In the present study, the 2D geometry is drawn within the software used [20]. The different sections of the model with varying material properties are drawn as separate regions. The pressure sensor positions along the dam body are defined as nodes within the numerical model to allow for juxtaposition of results from the numerical and physical modeling efforts.

3.2.1. Material Properties

Definition of the hydraulic properties is an important and challenging configuration of the model. The multiphase nature of throughflow with water displacing air in the voids requires consideration of saturation. When defining the model case, it can be considered a fully submerged case or a unsaturated/saturated case. The unsaturated/saturated case is selected for the present model. The water content and hydraulic conductivity can consequently be defined. Volumetric water content functions, are preinstalled in the professional software used in this study [20], based on grain size distribution and saturated water content. These parameters can be established by laboratory sieving analysis and porosity measurements. For the shell, the saturated water content is defined as 0.15, based on in-situ cone porosity measurements. From sieving analysis, the $d_{60}$ and $d_{10}$ were found to be 9 mm and 1.2 mm, respectively. The toe material grain size analysis yielded $d_{60}$ and $d_{10}$ to be 36 mm and 11 mm. Being coarser and uniformly graded, the toe material volumetric water content was estimated to be 0.4.

By using the non-Darcy add-in [23] the hydraulic conductivity function is replaced to replicate a non-Darcy condition. To enable this function, the add-in is selected as function type. For the shell, the saturated water content is set to 0.15, residual water content is set to 10% of saturated levels, 0.015, $\alpha$ is set to 8, $n$-parameter is set to 2, hydraulic conductivity is set to 0.003 m/s, the $l$-parameter is set to $-0.5$, the form-drag coefficient is set to 1.5 and finally temperature is defined as 20 $^\circ$ C. For the toe material the saturated water content is set to 0.4, residual water content is set to 0.04, again based on 10% of the saturated values. $\alpha$ is set to 15, $n$-parameter is set to 4, hydraulic conductivity is set to 0.1 m/s, the $l$-parameter is set to $-1$, the form-drag coefficient is set to 0.75 and temperature is again defined as 20 $^\circ$ C. A summary of the calibrated parameter set can be seen in Table 3.

**Table 3.** Summary of numerical model parameters.

| | Vol. Water Content | | Fitting Param. | | Hydr. Cond. | Tortuoisity & Connectivity | Form Drag Constant | Fluid Temp. |
|---|---|---|---|---|---|---|---|---|
| | Sat. | Res. | $\alpha$ | $n$ | $k$ (m/s) | $l$ | FDC | T (°C) |
| **Shell** | 0.150 | 0.015 | 8 | 2 | 0.003 | −0.5 | 1.50 | 20 |
| **Toe** | 0.40 | 0.040 | 15 | 4 | 0.100 | −1.0 | 0.75 | 20 |
| **Limit** | [0–1] | [0–1] | - | >1 | - | - | - | - |

### 3.2.2. Mesh

The automated mesh is drawn based on a global element size of 0.02 m, shown in Figure 7. This mesh size was selected as it provides good resolution and manageable calculation times. Specific nodes were placed to match the position of the pressure sensors, where node 2 and node 16 correspond to sensors P1 and P2, respectively. Nodes 5–11 correspond to sensors P3–P9, node 12 corresponds to sensor P10. However, for the numerical analysis, P1 is located outside the dam domain, which makes it unavailable for comparison. Similarly, for no toe and internal configuration, P10 is located outside the dam domain, meaning that values cannot be compared for these two toe configurations. Upon completion of the meshing, the analysis is ready to be run. The results produce the phreatic surface and pore pressure development through the dam. To compare results between the numerical analysis and the physical tests, data can be exported from the numerical model. The points of interest being the pore pressure development along the points matching the positions of the pressure sensors, the positions and corresponding sensors are listed in Table 1.

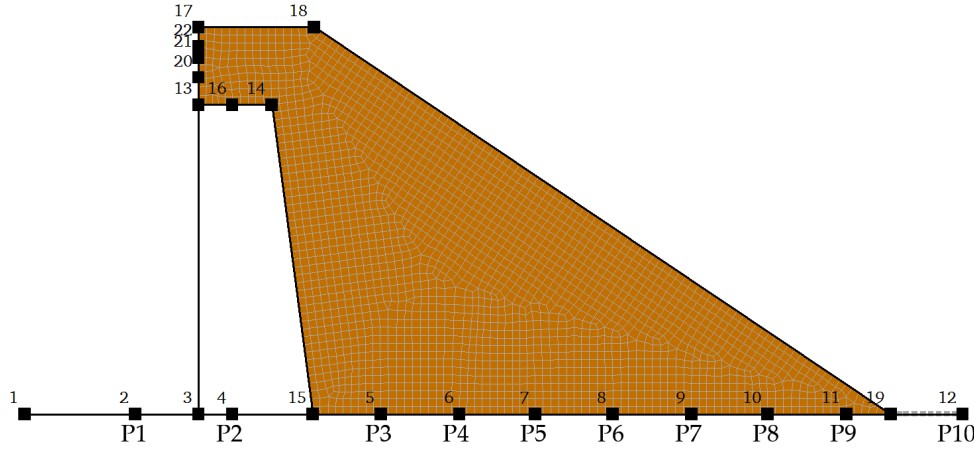

**Figure 7.** Design of the numerical model, showing the mesh and numbered nodes.

### 3.2.3. Boundary Conditions

To run a simulation, the boundary conditions must be defined. First, a total head or pressure head boundary must be added. In this case, a zero-pressure point is added to the downstream toe. Furthermore, the drainage boundary was set along the upper edge of the dam, set as a water-rate of 0 m³/s with potential seepage face review—this allows water to escape the domain upon reaching the boundary. Lastly, the input is defined through a water flux for each discharge level. The flux is entered as m³/s/m², which requires the correct area of flux to be defined. This is done by selecting the appropriate water level from the physical test results. For example, for an applied discharge of $q = 1 \times 10^{-3}$ m³/s, the average entry water level was measured at 0.87 m (P1) over the horizontal platform for all the tested models. The water flux is therefore defined as $1 \times 10^{-3}$ m³/s for the 0.07 m above the crest of the core. Similarly, for $q = 2 \times 10^{-3}$ m³/s the inflow occurs along 0.15 m of the face. Lastly, for $q = 4 \times 10^{-3}$ m³/s the inflow boundary is 0.2 m, the height of the crest. The setup is shown in Figure 8.

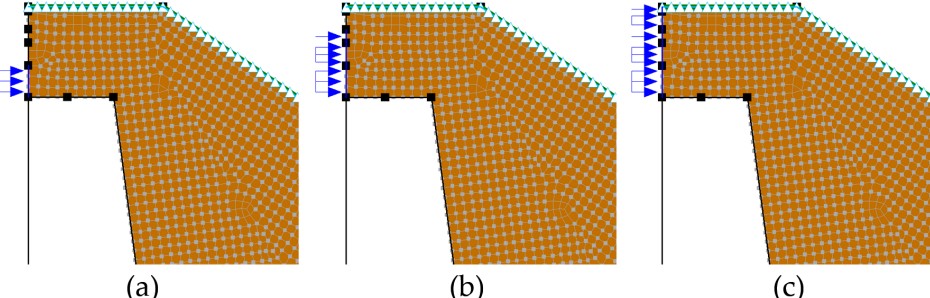

**Figure 8.** Input boundary flux shown for (**a**) $q = 1 \times 10^{-3}$ m$^3$/s, (**b**) $q = 2 \times 10^{-3}$ m$^3$/s and (**c**) $q = 4 \times 10^{-3}$ m$^3$/s.

### 3.2.4. Calibration and Evaluation

The parameter values that were selected for the numerical analysis were attained through trial and error. It was decided to utilize one parameter set which would provide the best fit for all configurations. Firstly, the results from numerical modeling of the no toe configuration were calibrated with the physical results. Upon arriving at a parameter-set, the parametric assumptions were further tested on the external, internal and combined configurations to evaluate the model performance. Necessary modifications/fine adjustments were made to the assumptions to achieve better fit with physical observations. The iteration process was repeated until an overall satisfactory fit was obtained. To evaluate the accuracy of the parameter-set, the main metric used was the root-mean-square error (RMSE), which calculates the standard deviation of the error between modelled and observed results. The root-mean-square value is favorable due to the resulting error being directly readable in the same unit as the modelled variable.

## 4. Results

The outcomes from the numerical modeling efforts to simulate observations and measurements from the physical modeling investigations are discussed within this section of the article. To enable visual comparison of the results, graphical plots of the dam models are depicted with pressure contours (pore pressure distributions) for incrementally applied discharge levels (Figure 9). The numerical and physical results are then collated through the comparison of the phreatic line for each individual discharge magnitude (Figure 10). Analysis of the modeling accuracies is further quantitatively described employing a statistical methodology. To aid in reading the results between numerical and physical models, the pressure sensors located along the dam body are given on the *x*-axis, with the positions provided in Table 1.

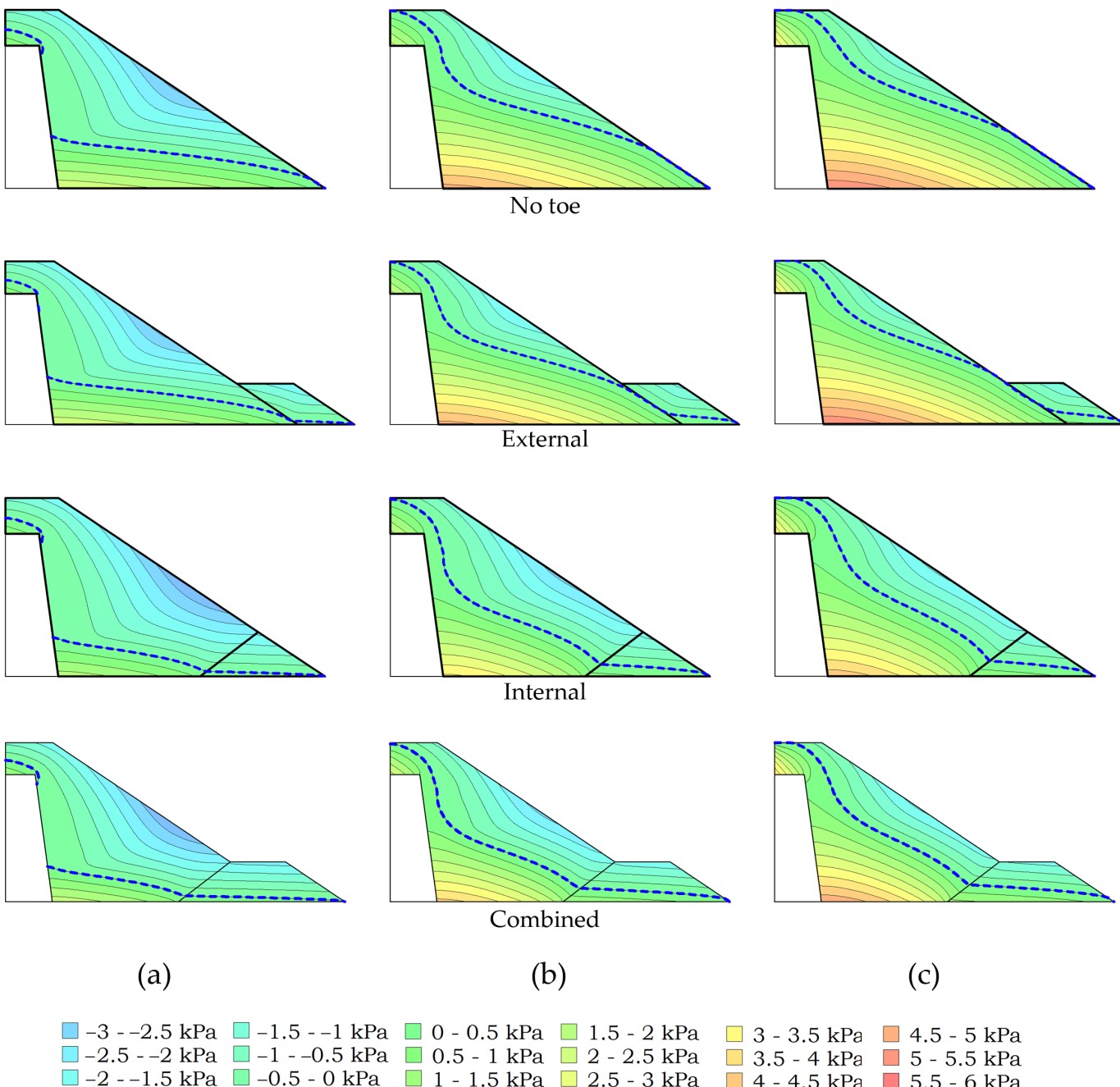

**Figure 9.** Numerical modeling results for the four toe configurations; column (**a**) shows results for $q = 1.0 \times 10^{-3}$ m³/s, column (**b**) $q = 2.0 \times 10^{-3}$ m³/s and column (**c**) $q = 4.0 \times 10^{-3}$ m³/s. The colored contours display the water pressure shown in the legend, while the phreatic surface is displayed as the dotted blue line.

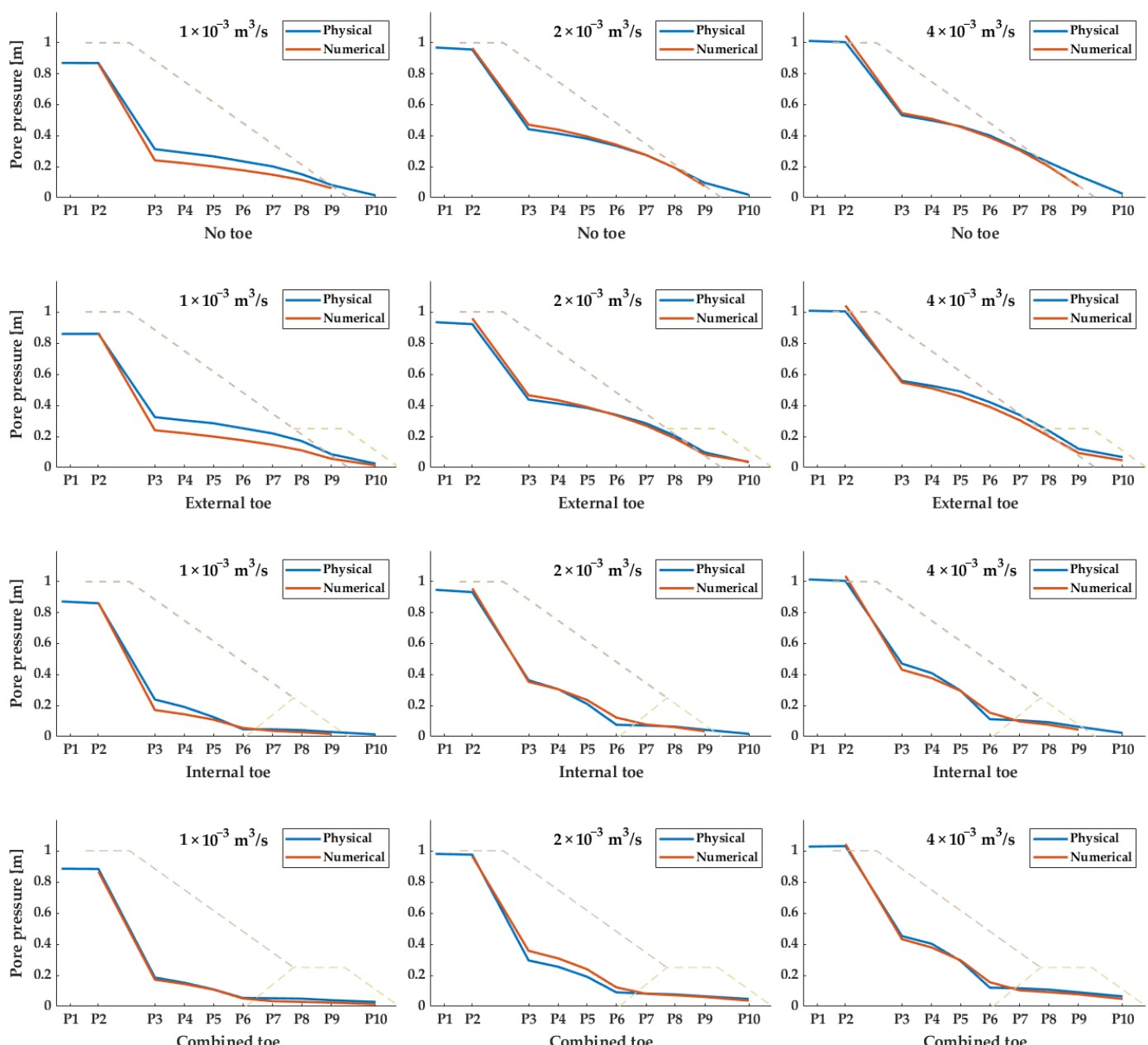

**Figure 10.** Numerical modeling results juxtaposed to physical modeling results. The stippled line displays the outline of the dam and toe configuration.

### 4.1. General Results from Numerical Modeling

The outcomes of the numerical analysis are summarized in Figure 9 as images of various model configurations as a function of the applied discharge. The pore pressure distributions are displayed as color contours with zero pressure contours marked as dotted blue lines representing the phreatic surface. In general, the calibrated numerical model demonstrates the good ability of the model to simulate throughflow conditions within the various dam models.

Results from the numerical analysis are overlaid with measurements from the physical modeling studies as depicted in Figure 10 for discharge magnitudes, $q = 1.0 \times 10^{-3}$ m$^3$/s, $2.0 \times 10^{-3}$ m$^3$/s and $4.0 \times 10^{-3}$ m$^3$/s. Juxtaposition of the results in Figure 10 illustrates that the numerical model accurately predicts the throughflow patterns within the various rockfill dam models subjected to various throughflow levels. The influence of the high permeability zones, i.e., the various toe configurations, on the development of phreatic

lines within the dam models is also accurately modelled. The internal and the combined toe configurations lead to reduced pore pressure levels within the dam structure for a given applied throughflow magnitude in comparison with rockfill dam models with no toe. Furthermore, the external toe configuration appears to have minimal impact on throughflow development within the dam structure. Both these findings are in alignment with the documented findings from the physical studies [13]. The numerical model also appears to be capable of simulating the flow transitions from (a) the dam core towards the shoulder and (b) from the downstream shoulder towards the toe structure. This was verified by visual inspection of the video footage from the physical tests. Further detailed descriptions of qualitative results from comparative evaluations conducted between various physical and numerical model configurations are detailed in subsequent sections.

### 4.1.1. No Toe

In the depictions of Figure 10 for the no toe model, the phreatic surfaces undergo transitions as they enter the downstream shoulder structure. The numerical simulations appear to closely resemble the physical observations in this regard. Further, the precited pore pressure development trends in general confirm well the observations for sensor locations P2 to P8. From P8 to P9, the numerical predictions follow the dam surface, which is defined as a drainage boundary. In the numerical model, the zero-pressure boundary point is located at the toe, whereas in the physical model the water column at the downstream end of the dam, resulting from the throughflow exiting the dam, is measured. This leads to divergence in the results at the downstream end of the dam structure.

### 4.1.2. External Toe

The comparative evaluation results for rockfill dams with external toe configuration are shown in Figure 10. The pressure developments through the dam structure, as previously stated, appear very similar to the results from simulations for dams without toe structures. Comparison of measurements from sensor P10 with the simulations demonstrates good fit for flow within the external toe structure.

### 4.1.3. Internal Toe

Results for the internal toe configuration can be seen in Figure 9. The phreatic line can be seen to undergo gradual decrements along the dam length and experiences marked lowering of pore pressure prior to entering the toe structure. Furthermore, the phreatic surface within the toe structure drops significantly and flattens towards the exit boundary. Figure 10 shows a comparison of the physical and numerical results. The results in general correlate well with each other as documented by the depictions for the internal toe. Within the toe structure, the two results have a similar trajectory, however the numerical model analysis translates towards zero at the end of the toe (defined boundary condition), whereas the physical measurement is slightly higher due to the throughflow water exiting the dam.

### 4.1.4. Combined Toe

The numerical results in Figure 9, for dam model coupled with the combined toe configuration, closely resemble the results from the internal configuration case. However, with a longer flat decrease in pore pressures within the toe. For the combined toe configuration, a near perfect fit is obtained to the physical model results for $q = 1.0 \times 10^{-3}$ m$^3$/s.

### 4.2. Performance Evaluation

This section aims at quantitative evaluation of the model performance as pertained to ability of the numerical model to simulate phreatic surface developments within the model dam structures. To accomplish this task, a statistical evaluation is conducted adopting a relative changes approach. Relative changes are computed adopting the methodology put forth by Kiplesund et al. [13], wherein percentage pore pressure differences were computed

for the different models with the no toe case as the baseline. Similar computations were conducted here for results from the numerical modeling investigations producing Table 4. This gives a good picture of quantitative conformity between results from the numerical and physical models.

Table 4 does not include P9 due to the boundary conditions in place, as the phreatic line of the numerical model will automatically follow the dam surface and thus is not comparable with measurements at location P9 in the physical models. The table in general demonstrates conformity between the percentage pressure changes with the physical modeling results. Hence, the numerical model is able to accurately simulate the pressure profiles at various locations within the dam structure. The model also accounts for the influence of the high permeability zones to a good degree.

The root-mean-squared error (RMSE) was further calculated for each model configuration exposed to various discharge magnitudes. The results show that the RMSE in general increases with increasing discharge. The cumulative RMSE for all models was found to be 0.023 m, and the largest error is computed for $q = 1.0 \times 10^{-3}$ m$^3$/s.

**Table 4.** Comparison of pressure reduction for different toe configurations relative to no toe configuration for physical and numerical models.

| Toe Config. | $q$ (L/s) | Physical Model Rel. Pressure Reduction (%) | | | | | | | Numerical Model Rel. Pressure Reduction (%) | | | | | | |
|---|---|---|---|---|---|---|---|---|---|---|---|---|---|---|---|
| | | P2 | P3 | P4 | P5 | P6 | P7 | P8 | P2 | P3 | P4 | P5 | P6 | P7 | P8 |
| External | 1.0 | −1 | 3 | 4 | 7 | 8 | 9 | 12 | 0 | −1 | −1 | −1 | −1 | −2 | −3 |
| | 1.5 | −2 | 2 | 2 | 3 | 4 | 5 | 7 | 0 | −1 | −1 | −1 | −1 | −2 | −3 |
| | 2.0 | −4 | −1 | −1 | 1 | 1 | 3 | 6 | −1 | −1 | −2 | −2 | −2 | −2 | −3 |
| | 2.5 | −3 | 0 | 1 | 2 | 3 | 5 | 8 | 0 | 0 | 0 | −1 | −1 | −1 | −1 |
| | 3.0 | −2 | 1 | 2 | 3 | 4 | 6 | 7 | 0 | 0 | 0 | 0 | 0 | 0 | 0 |
| | 3.5 | −1 | 5 | 6 | 6 | 5 | 8 | 4 | 0 | 0 | 0 | 0 | 0 | 0 | 1 |
| | 4.0 | 0 | 5 | 5 | 6 | 5 | 8 | 4 | −1 | 0 | 0 | 0 | 0 | 0 | 0 |
| Internal | 1.0 | −1 | −24 | −34 | −53 | −80 | −77 | −73 | 0 | −29 | −36 | −46 | −68 | −75 | −75 |
| | 1.5 | −1 | −17 | −26 | −46 | −78 | −75 | −69 | −1 | −27 | −33 | −43 | −66 | −73 | −72 |
| | 2.0 | −3 | −18 | −26 | −44 | −77 | −74 | −67 | −1 | −25 | −31 | −40 | −64 | −72 | −69 |
| | 2.5 | −1 | −16 | −24 | −42 | −76 | −72 | −64 | 0 | −20 | −26 | −35 | −59 | −66 | −59 |
| | 3.0 | 0 | −14 | −21 | −39 | −74 | −70 | −61 | −1 | −22 | −27 | −37 | −62 | −70 | −65 |
| | 3.5 | 0 | −11 | −18 | −36 | −73 | −68 | −61 | −1 | −22 | −27 | −36 | −61 | −69 | −64 |
| | 4.0 | 0 | −11 | −18 | −36 | −72 | −67 | −59 | −1 | −21 | −26 | −35 | −60 | −68 | −62 |
| Combined | 1.0 | 2 | −41 | −48 | −59 | −77 | −74 | −67 | 0 | −29 | −35 | −46 | −71 | −77 | −74 |
| | 1.5 | 2 | −35 | −41 | −53 | −75 | −72 | −64 | 0 | −26 | −32 | −43 | −67 | −74 | −68 |
| | 2.0 | 2 | −33 | −39 | −50 | −73 | −70 | −60 | 0 | −24 | −30 | −40 | −64 | −71 | −63 |
| | 2.5 | 2 | −32 | −36 | −47 | −71 | −68 | −56 | 0 | −20 | −25 | −35 | −60 | −65 | −54 |
| | 3.0 | 2 | −26 | −31 | −43 | −69 | −65 | −53 | 0 | −22 | −27 | −37 | −62 | −68 | −58 |
| | 3.5 | 3 | −15 | −20 | −39 | −71 | −64 | −54 | 0 | −21 | −26 | −36 | −61 | −67 | −56 |
| | 4.0 | 3 | −15 | −19 | −36 | −70 | −63 | −53 | 0 | −21 | −26 | −35 | −60 | −66 | −55 |

| Legend | 0–20% | 21–40% | 41–60% | 61–80% |
|---|---|---|---|---|

### 4.3. Laminar versus Turbulent Flow in Numerical Models

The numerical results presented are all models that consider a non-linear flow regime through the use of the non-Darcy tool [23]. However, the traditional seepage modeling, considering laminar flow, was also investigated. It was found that the seepage through the shell material, representing the rockfill shoulder, could be reliably modelled in a laminar regime for the no toe model. Difficulties occurred, in models with a toe configurations, when the flow transitioned into the toe region. Firstly, there were convergence issue with the analysis.

Secondly, to achieve comparable results to the non-Darcy modeling, it was necessary to alter the material parameter set between the models of different toe configurations.

## 5. Discussion

The results from the numerical analysis demonstrate that turbulent non-Darcy flow through rockfill dam structures can be modelled with good calibration metrics. However, some challenges with the numerical modeling work were encountered. This section discusses these challenges and aims at putting forth recommendations and insights that can potentially supplement further research in this regard.

### 5.1. Boundaries

The upstream boundary condition is simplified by assuming that the discharge is evenly distributed along the corresponding water level (see Figure 8). Thus, the velocity profile will be homogeneously distributed along the face. This can be stated as a simplification. However, due to the low entry velocities at the entry surface, the variability in the velocities with depth (0.2 m high crest) can be considered as insignificant. Additionally, the effect of this simplification on the results of the study are deemed to be minimal.

Definition of a drainage boundary with a potential seepage face results in that water is able to escape the domain along the boundary. In the numerical model, the dam surface is a drainage boundary, including the surface at the crest. Thus, for the high upstream phreatic line, as occurs for the highest discharges, a minor amount of water exits the dam at the crest where the inflow enters the dam, as shown in Figure 11. The effect was observed in the numerical model results for discharges, $q = 2.5 \times 10^{-3}$ m$^3$/s and higher. To ensure that drainage out of the system does not distort the results in the downstream dam shoulder, data were extracted to calculate the water loss at the highest applied discharge, $q = 4.0 \times 10^{-3}$ m$^3$/s. Considering the external toe case as an example, the total loss was computed to be $5.7 \times 10^{-8}$ m$^3$/s. Furthermore, for the no toe configuration, the total loss was calculated as $6.8 \times 10^{-8}$ m$^3$/s. The magnitude of losses was therefore deemed negligible for all models. This further entails that the same applies to lower applied discharges, $q < 4.0 \times 10^{-3}$ m$^3$/s.

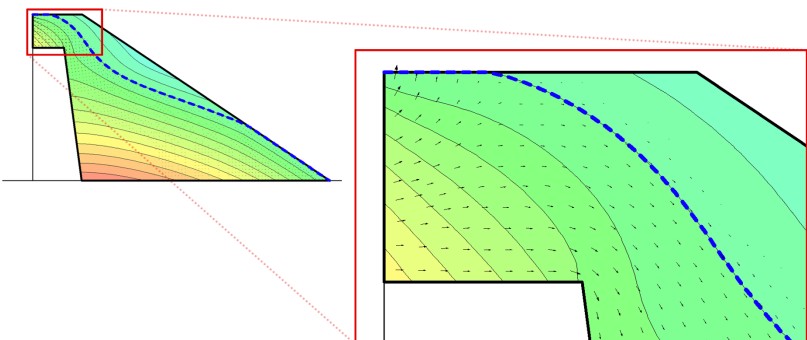

**Figure 11.** Detail showing flux vectors at the crest of the no toe configuration for $q = 4.0 \times 10^{-3}$ m$^3$/s. The maximum flux shown in the top left corner is 0.0068 m$^3$/s/m$^2$; the vectors are magnified by a factor of two to increase visibility.

### 5.2. Pressure Development

Investigating the pressure development within the dam models, for $q = 1.0 \times 10^{-3}$ m$^3$/s, the numerical results for the no toe and external toe models show somewhat lower pressure values than physical modeling observations. The reasoning for this could be that the selected parameter set and successive numerical results were better fitted to the higher discharges. However, for both the internal toe and combined toe cases, the fit is similar for all the discharges, and even best for $q = 1.0 \times 10^{-3}$ m$^3$/s in the case of the combined toe. Furthermore, the observed discrepancy does not affect the general outcome of the study relating to investigating the different toe configurations. Moreover, it is of value to be able

to use the same material properties for all the models and discharges. Comparison of the numerical and physical model results in Figure 10 is a validation of the parameter set used and supports further investigations using the numerical model.

The numerical results for the internal and combined toe configurations are in good agreement with the physical results. However, for location P6, close to the interface of the rockfill material and the toe, the pore pressure is slightly lower for the physical models for discharges $q = 2.0 \times 10^{-3}$ m$^3$/s and larger. This can be due to increased permeability at this location in the physical model, explained by the methodology adopted for construction of the dam. The toe is first placed in its position in layers and the dam is built adjacent to the toe. When compacting the dam layers, the shell material in contact with the toe needs to be carefully tamped, so as to not cause filling of the voids or disturbing the shape of the toe. This can cause the interface and surrounding area to have greater permeability than the rest of the dam. For the numerical model, the material properties are defined for the respective region, and the interface has no effect outside of the contact. A similar limitation lies at the crest of the dam, which can be seen as the numerical results yielding higher pressure values at P2 for increasing discharge. Construction of the physical model requires a metal mesh at the face in order to withhold the materials from sliding into the upstream reservoir. This causes some of the finer material to slide through, as well as diminishing the compactability. Ultimately this can have an increasing effect on the permeability compared to the main body of the shell, which is not represented in the numerical model, as it remains completely homogeneous. These effects are important to note when proceeding to real dam cases, considering that the method of construction introduces regions of different permeability.

From Table 4, the physical model results show the relative reduction in pore pressure decreases with increasing discharge. The same overall trend can be seen for the numerical results. However, the results for $q = 2.5 \times 10^{-3}$ m$^3$/s, do not agree with that trend. These peculiarities can be linked to issues with the input boundary. For $q = 2.5 \times 10^{-3}$ m$^3$/s, the water level was measured at 0.99 m at P1, meaning that the mesh size used, of 0.02 m, is split for this edge.

A consequence of lowering the phreatic surface, due to introduction of a high permeability zone such as the toe installations, is increased flow velocities within the dam structure. Increased velocity can have a detrimental effect on stability through internal erosion processes, if it is not accounted for in the dam design. To investigate the velocity increase, the numerical model was used, extracting data at a horizontal line from the top of the core into the dam. Comparing the internal toe configuration to no toe, the data showed a 10% average increase in velocity fluxes for $q = 4.0 \times 10^{-3}$ m$^3$/s.

*5.3. Calibration*

The parameter set that was obtained for the dam and the toe structure was arrived at through trial and error. When using the non-Darcy add-in, there are multiple physical properties and fitting parameters that affect the flow patterns and subsequent phreatic line developments. The critical component being the hydraulic conductivity, defining the permeability of the dam. Minute adjustments of the hydraulic conductivity will have great influence on the phreatic line and pore pressure development. Upon setting an agreeable hydraulic conductivity, the Van Genuchten fitting parameters and tortuosity parameters were dialed in.

The $\alpha$-parameter will alter the fit at the upstream end of the dam, showing influence on the early sensor locations. It was found that increasing the $\alpha$-value in general raised the phreatic surfaces.

The *l*-parameter had significant influence on the flow pattern through the downstream dam shell and the toe structure. It was found that, when decreasing the *l*-parameter flow, velocity vectors were seen to a larger degree above the phreatic lines. This effect has the largest influence on the toe, where there is high inter-connectivity of the voids. The *l*-parameter was also found to affect the pressure in the main body of the dam.

Some limitations on the parameters are hard to estimate, as the user interface of the non-Darcy add-in of the software used [20] does not limit any input, but incorrect values will cause failure to find a solution during analysis. For example, the *n*-parameter is said to have a limit yet, with increasing hydraulic conductivity, the limit changes. In this sense, the add-in calibration can be slightly cryptic, prolonging the process. The calibration process aimed at finding a parameter set best fitting P2-P10, as the phreatic lines on top of the core remained largely unaffected by toe design in the physical model. However, for internal and no toe configurations, the last measurement points, P9 and P10 were affected by the end of the domain.

In examining the available literature regarding the Van Genuchten input parameters, it should be clarified that the described parameters are detailed for soils of different compositions. Another limitation to bring up is that the apparent conductivity used in the non-Darcy add-in is designed for groundwater aquifers, and is valid when velocities remain low to intermediate [23]. This could pose issues with upscaling the model to larger dams where velocities can be considerably larger. There is no available research utilizing the non-Darcy add-in, which further adds some uncertainty.

Equifinality of parameters is an additional point of discussion for the non-Darcy modeling. As calibration is done on a trial and error basis, with multiple fitting parameters, the results could possibly be reproduced with another parameter-set.

*5.4. Application and Future Recommendations*

The study demonstrates how numerical models can be useful for deeper apprehension of the results from physical tests. The numerical model enables detailed investigation of flow through the dam structure at every specific location, not just discreet positions determined by, e.g., installed pressure sensor locations. Moreover, the numerical model has the advantageous possibility of investigating different parameters at specific locations that the physical test cannot, such as velocity. In addition, through a calibrated numerical model, one can experiment with modifications to the physical model which are more resource intensive than modifications to numerical models. Hence, prior to customization of the physical model, it is highly recommended to utilize the numerical model for planning of future experiments. The numerical model requires very little resources for alteration of the design, which can then be used to design the changes and hypothesize the results of alterations to the physical model.

It is important to proceed from physical scale models to full scale dam cases, preferably with relevant data for calibration and validation. In general, to optimize the calibration process, it is recommended to investigate the usage of optimization algorithms to find optimally fitted parameter sets. With adoption of an optimization algorithm or machine learning method, higher precision calibrations could be achieved, but one must be cautious of unrealistic parameters. Before expanding the numerical model to other cases, it is recommended to verify the numerical model on a prototype rockfill dam with well-documented throughflow data. Calibrating the model to a full-scale embankment dam can provide verification of the applicability and validity of the non-Darcy add-in within prototype dams.

## 6. Conclusions

The goal of this study was to investigate the applicability of a professional software to numerically model turbulent flow through rockfill dams. Moreover, it was to further understand the effects of different rockfill toe configurations within the downstream dam shoulder. Numerical models were successfully calibrated against results from physical model tests, employing one set of material parameters for different model setups. Through this, the present article makes a strong case highlighting the potential for numerical modeling of turbulent non-Darcy flow through rockfill dam structures. The numerical analysis results further support findings of the physical study [13] relating to effectiveness of the different toe configurations. In comparison to the dam without a toe, the external toe

protects the exit zone on the downstream side from eroding supporting the findings and recommendations of Moran et al. [11]. Additionally, the internal and combined toe configurations are effective in lowering the phreatic line within the dam, for enhanced slope stability compared to the cases without a toe or an external toe.

The numerical modeling study presented demonstrates the efficacy of the model with regard to predicting throughflow in rockfill dams to a high degree of accuracy. Numerical modeling tools over the years have become increasingly reliable and robust and the trend appears to extend into the future. Development and calibration of numerical modeling tools to assist in the design and evaluation of rockfill dams can be stated as an effective method for enhancing dam safety. Since a significant number of iterations can be run in such a numerical model, a wide variety of material properties and loading conditions could be evaluated leading to better practical decision-making. Such numerical models can also be invaluable for research and development. A small number of physical modeling studies can lead to reliably calibrated numerical models and these models can further be employed to carry out a spectrum of investigations on model variations. Further research into development, calibration and, in turn, validation of numerical modeling tools within the research discipline of rockfill dam engineering is highly recommended.

**Author Contributions:** Conceptualization: F.G.S., G.H.R.R.; methodology: N.S.S., G.H.R.R., F.G.S.; formal analysis: N.S.S.; investigation: N.S.S., G.H.R.R., F.G.S.; original draft preparation: N.S.S., F.G.S., G.H.R.R.; writing: N.S.S., F.G.S., G.H.R.R.; visualization: N.S.S.; supervision: F.G.S.; project administration: F.G.S. All authors have read and agreed to the published version of the manuscript.

**Funding:** Financial support for the research venture was provided by Hydrocen, Norway.

**Institutional Review Board Statement:** Not applicable

**Informed Consent Statement:** Not applicable.

**Data Availability Statement:** Data presented in this study can be made available upon request from the corresponding author.

**Acknowledgments:** The authors acknowledge the support and co-operation offered by Geir H. Kiplesund, NTNU and Marius Rokstad, NTNU with this research project. Appreciation also goes to Livia Pitorac for allowing the use of her photo (real dam cases) within Figure 3.

**Conflicts of Interest:** The authors declare no conflict of interest.

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
