# Peer review of "Numerical Modeling of the Effects of Toe Configuration on Throughflow in Rockfill Dams"

_water, doi:10.3390/w13131726_

Round 1

Reviewer 1 Report

This work applied a commercial software to numerically model investigating on the throughflow in dams with different rockfill toe configurations. However, there are still some details that need to be further clarified and supplemented by the authors.

  1. Line 32: what dose the word “ICOLD” mean? Please provide the full name when it first appears, or give some explanation.
  2. Line 149: “Several past studies ……” Please list serval references as examples after this sentence.
  3. It’s better to put section 3.2.3 (Mesh) before section 3.2.2 (Boundary conditions).
  4. Line 239: “the l-parameter is set to -1”, the variable l should be italic.
  5. Line 280: the expression of “the root-mean-square method (RMSE)” is different from the expression in later: like Line 371 “The root-mean-squared error (RMSE)”.
  6. Please introduce more details about the commercial software used in this work, for example, the governing equations solved, the numerical scheme, and so on.
  7. Please mark the sensors’ position out in Figure 8.
  8. There are flow fields drawn in Figure 9, but it can not be seen clearly, please modify the figures. In addition, the scale of velocity arrows should be specified.

Author Response

Dear Reviewer,

Please see the attachment for our response to the comments. We've included both reviewers' comments in the response, so that all changes are made clear.

Thank you very much for you time and contribution!

Kind regards,

Nils Solheim Smith

Reviewer 2 Report

The reviewer wants to thank the authors for their paper presenting a numerical study reproducing the experimental investigation of the throughflow in a rockfill dam. S/he has some (minor) comments and questions:

*1) Abstract: line (L) 4: The reviewer would suggest being very precise here and clarify that only the core is overtopped. In a first moment, it could also be interpreted that the full dam is overtopped.

*2) L149: please provide the specific references to those mentioned past studies.

*3) Table 1/Figure 5: please introduce the coordinate system and especially the origin.

*4) Section 3.2.2: a) The later used 4*10^-3 m^3/s is not introduced. b) Please provide for all three discharges the pressure level at P1, which leads to the chosen inlet height. Is this realistic that a homogeneous velocity profile is assumed? c) the reviewer is not fully sure, but s/he assumes that in the experimental investigation the discharge was fixed and the height variable, correct? Would it be possible to define a water height pressure in the numerical model and verify that the same discharge enters the section? d) Table 4 presents the results for 7 different discharges and again it would be good to understand the chosen boundary condition for each one and understand how sensitive the results are based on this chosen (??) value.

*5) L304: Here three discharges are used but previously only two were introduced. See previous point.

*6) Figure 10: Maybe in addition to the point *4, the bigger discharges indicate a higher pressure at the section P2 for some cases. The reviewer might miss the discussion of this and apologise in case.

*7) Section Calibration and general: Please allow the reviewer to be over critical. The numerical simulation can reproduce the pressure distribution based on a specific calibration. Ok, that is good news but doesn’t bring a very big novelty hence all those values are known. The reviewer understands the usage of the numerical simulation in future, but s/he has the feeling that the authors are very careful not to mention the result of the calibration of the numerical values. Based on this, the reader could get the feeling that the values had to be changed to unrealistic values, to achieve the result. The reviewer doesn’t hope that this is true but would highly recommend that the authors provide an overview of the tuned values and the range. It would be very beneficial for future usage of the software.

Thank you very much and the reviewer is looking forward reading the paper again.

Author Response

(The authors gave the same response as above.)

Round 2

Reviewer 2 Report

The reviewer thanks the authors for their answers and corrections. One small thing: Please add, x to the table 1 in addition or instead of position, so that this is clear. Thank you. The reviewer is looking forward to the publication.